# Dementia is our "biggest expanding caseload": Core learning for student speech and language therapists

Anna Volkmer[1]*, Reem S.W. Alyahya[2], Hannah P. Atkinson[3], Arpita Bose[4], Hannah Britton[5], Lindsey Collins[6], Katie Earing[7], James Farraday[8], Dharinee Hansjee[9], Mary Heritage[10], Sophie Mackenzie[11], Sally Pratten[12], Catherine Tattersall[13], Daniel Underdown[14], Allison Virgilio[15], Nicola Withford-Eaton[16], Mirjam Gauch[17], Richard Talbot[1], Jackie Kindell[18]

1 Department of Psychology and Language Sciences, UCL, London, United Kingdom, 2 Department of Language and Communication Science, School of Health and Medical Sciences, City St George's, University of London, London, United Kingdom, 3 Speech & Language Therapy Division, School of Health and Social Care, University of Essex, Essex, United Kingdom, 4 School of Psychology and Clinical Language Sciences, University of Reading, Reading, United Kingdom, 5 Department of Speech and Language Therapy, Queen Margaret University, Edinburgh, Edinburgh, Scotland, United Kingdom, 6 Centre for Applied Dementia Studies, University of Bradford, Bradford, United Kingdom, 7 Cardiff School of Sport and Health Sciences, Cardiff Metropolitan University, Cardiff, United Kingdom, 8 The Newcastle upon Tyne Hospitals NHS Foundation Trust, Newcastle, United Kingdom, 9 School of Health Sciences, University of Greenwich, London, United Kingdom, 10 Speech and Language Therapy Department, University of Lincoln, Lincoln, United Kingdom, 11 School of Health and Rehabilitation Sciences, Health Sciences University, Bournemouth, United Kingdom, 12 Department of Health Professions, Manchester Metropolitan University, Manchester, United Kingdom, 13 School of Allied Health Professions, Nursing and Midwifery, University of Sheffield, Sheffield, United Kingdom, 14 School of Sport, Health and Wellbeing, Plymouth Marjon University, Plymouth, United Kingdom, 15 School of Human and Health Sciences, Univeristy of Huddersfield, Huddersfield, United Kingdom, 16 Department of Health and Care Professions, Birmingham City University, Birmingham, United Kingdom, 17 Department of Psychiatry and Psychotherapy, University Medical Center of the Johannes Gutenberg University Mainz, Mainz, Germany, 18 Division of Psychology, Human Communication and Neuroscience, University of Manchester, Manchester, United Kingdom

* a.volkmer.15@ucl.ac.uk

## Abstract

Dementia is a relatively recent addition to the speech and language therapist's clinical role. Given the increase in prevalence of dementia, a review of current student speech and language therapy training on this topic is essential to ensure the profession can meet the needs of this expanding population. This study therefore aimed to understand the current support and training being provided for pre-registration student speech and language therapists on the topic of dementia across UK universities and explore the experiences and views of lecturers delivering this training. This study used an explanatory sequential mixed-methods study, employing a survey which consequently informed two focus groups. In total 18 participants from 16 universities participated in the study. Reflexive thematic analysis identified six main themes in the focus group data; 1. Dementia is a vast and therefore complex topic, 2. There are biases about dementia within and outside the profession, 3. Students bias towards

**Data availability statement:** The datasets generated and analysed during the current study are not publicly available in their entirety due to the terms of our ethical approval. Extracts can be made available on request by emailing the UCL ethics committee on ethics@ucl.ac.uk.

**Funding:** The author(s) received no specific funding for this work.

**Competing interests:** The authors have declared that no competing interests exist.

dementia can be shifted through exposure, 4. Teaching could be enhanced by threading dementia through the curriculum, 5. There are several tensions in teaching on dementia: Possibilities versus clinical realities now and in the future, and 6. Dementia teaching must focus on person centeredness. The results emphasise the need for a paradigm shift in the teaching of student speech and language therapists. Underpinned by the principles of person-centered care five core components for teaching dementia student speech and language therapists were synthesized comprising 1. Challenge stereotypes around dementia, 2. Focus on speech, language and communication across dementias, 3. Teach them to build a relationship with people affected by dementia, 4. Teach goal setting for a progressive trajectory, 5. Prepare them to advocate for gold standard. Future research should work with people with dementia to further refine the core components for teaching student speech and language therapists.

## Introduction

Worldwide the number of people living with dementia is rising [1]. The number of people living with dementia in the UK is currently estimated at around 944,000 and is set to increase to 1.2 million by the year 2030 [2]. People over the age of 55 are now more afraid of being diagnosed with dementia than other conditions such as cancer [3,4]. This fear is reinforced by therapeutic nihilism [5] and stereotypical depictions of people with dementia [6,7].

Whilst research has endeavored to develop disease modifying interventions, with some significant recent advances in the field of Alzheimer's diseases [8], there are currently no widely available curative treatments for dementia. There are however a range of options to support people to live well with their dementia, such as pharmacological, behavioural and social interventions that aim to improve quality of life, maintain cognitive strengths, involve people in decisions about their care and enable them to participate in activities of daily living [9]. However, within speech and language therapy there is sometimes a lack of awareness about what can be done to help people with dementia and their families, possibly because this is a developing area of practice for the profession [10]. Indeed, research on speech- language interventions in language led-dementias has significantly increased in number and rigour in the last decade [11]. A systematic review published in 2013 identified 39 articles, predominantly focused on single case studies [12]. This has expanded to 103 studies in 2023 [11], with several group trials and two recent randomized controlled trials [13,14].

Dementia care is also a relatively recent addition to the speech and language therapy training curriculum. In the UK context speech and language therapists (SLTs) provide treatment, support, and care for people of all ages who have difficulties with speech, language, communication, eating, drinking and swallowing [15]. The profession has expanded over time into different fields and client groups, from an early focus on speech and language therapy with children, then to adults with aphasia and other acquired communication difficulties, and more recently to include management

of swallowing difficulties [16]. In terms of adult acquired communication difficulties there is a long-standing tradition of SLTs working with people with stroke aphasia using didactic therapy approaches that begins prior to World War II [17]. Progressive neurological conditions such as motor neurone disease, Parkinson's and dementia, have been a later addition to the SLT role and training [18]. Publications began to argue for the SLT role in the 1990s [19] and the first position paper on dementia practice in SLT was published as recently as 2005 [20]. In fact, it is the position of the Royal College of Speech and Language Therapists (RCSLT) that SLTs have a key role in the diagnosis and treatment of speech, language and communication difficulties in dementia [20,21]. Training carers to help them adapt their communication style to support and improve interactions with the person living with dementia is advised as essential by both RCSLT [21] and National Institute for Health and Care Excellence [9]. Speech and language therapy training therefore now includes developmental and acquired difficulties across the full lifespan, ranging from dysphagia practice in neonatal care to issues within ageing and end of life.

Proportionately, it is likely that the current speech and language therapy curriculum needs to focus even more on dementia. People with dementia have been reported as one of the biggest and most rapidly expanding areas of practice for SLTs [22,23], and indeed increasing referral numbers have been reported in Australia [24], the UK [25], Italy [26] and Turkey [27]. Yet within the UK, clinical SLTs report a lack of confidence and knowledge around working with people with dementia-related communication difficulties [10]. Volkmer's survey [10] highlighted that limited funding and restrictive service criteria prioritise swallowing difficulties and many services are unable to support people with dementia to address communication difficulties. This means that access to speech and language therapy support for communication difficulties related to dementia is variable across the country. Importantly, however, people with dementia and their family members living in community settings have identified conversations, communication and relationships as a key priority area for them [28,29]. Whilst the research evidence in this field is still developing, it seems logical that supporting communication to maintain independence, could also keep people at home for longer and reduce social and financial burdens.

Internationally, there is a recognized need to enhance the skills of SLTs to enable them to meet the needs of people with dementia and their families [30,31]. Beber et al. [32] highlight that, similar to the UK, many SLTs in Brazil do not feel confident in their own skills in working with people with dementia. More clinical experience working with people with dementia was associated with greater confidence, and respondents advocated for improved clinical guidelines and increased allocation of pre-registration training time on dementia [32]. In the USA Mahendra et al. [22] describe the development of their teaching using the three stage Awareness-Application-Advocacy Model. This includes personal reflection tasks on personal biases and knowledge gaps, developing knowledge and skills in speech-language assessment and management approaches, and developing competence through applied service-learning projects embedded in long-term care settings [22].

Whilst clinical placements are a core component of preregistration UK speech and language therapy courses, Banerjee et al. [33] have highlighted that these types of placements may be too short to allow health professionals to develop appropriate skills for working with people with dementia. Their scoping review of dementia-focused integrated clerkships (speciality clinical training course) and mentorship programmes for health professional training identified eight studies across the USA and Italy. They reported improvements in both knowledge, empathy and compassion amongst students. This informed the development of their own programme called "time for dementia" for student medics, paramedics and nurses in southern England [33]. This novel programme involves partnering students with a person with dementia, whom they visit for two hours every three months over a period of two years. The researchers concluded that the programme promoted a person centered approach and improved social comfort in students [33] as well as impacting on communication and well-being amongst the people with dementia [34]. Banerjee et al. [33] suggested that effective care for individuals with long-term conditions requires a different set of proficiencies to those with acute conditions, and therefore a different approach to clinical training across health care professionals is needed.

Given the need to continually review the curriculum to ensure it meets the needs of modern health and social care, it is important to review SLT's training curriculum across the UK to ensure it addresses the needs of people with dementia. Lecturers who are most familiar with, and who are required to deliver the teaching, are best placed to review the current

 

content. This study therefore aimed to explore the views of lecturers teaching student SLTs across the UK to answer the following research questions:

1. What support and training are currently being provided for pre-registration student SLTs on the topic of dementia?

2. What are the experiences of the lecturers in delivering this training?

3. What do lecturers think should be taught to improve confidence and competence in student SLTs to enable them to work with people with dementia?

## Materials and methods

This was an explanatory sequential mixed-methods study [35], employing a survey which consequently informed two focus groups. The survey allowed for collection of a breadth of data such as hours of teaching and topics taught and addressed the first research question. This informed the focus group topic guide that facilitated the in-depth exploration of experiences and opinions of staff engaged in the teaching of dementia, addressing research questions two and three. The online focus groups are described in line with online focus group reporting guidance [36,37]. This study was conducted in line with Mixed Methods Appraisal Tool guidance [38], a brief supplementary S1 File (1) showing item-by-item judgments has been included.

This study has been approved by the Chair of UCL Language and Cognition Ethics, project ID: LCD-2024-05. All participants provided informed written consent to participate in the study.

## Participants

The Royal College of Speech and Language Therapists (RCSLT) lists 25 accredited Higher Education Institutes (HEIs) approved by the Health and Care Professions Council to train SLTs in the UK. These HEIs offer a range of programmes at pre-registration level including undergraduate, masters level and apprenticeship courses. This list was used to tabulate the current HEIs and courses in the UK. The Committee of Representatives of Education in Speech and Language Therapy, which has representation from all HEI SLT program directors, was contacted on 23rd April 2024 via the RCSLT and asked to circulate an email to committee members, with a request for information about the study to be forwarded to staff teaching dementia content. Following this several staff contacted the researchers. Over the second week of June 2024 further information was sent directly to HEIs via the generic university email addresses, located from university websites, that had not yet been in contact. Six representatives from HEIs responded explaining that participation was not possible due to current time constraints, or content was taught by outside clinical staff who were not able to attend. By the end of June 2024 twenty-four representatives across 19 HEIs had expressed an interest in participating in the study.

Prior to attending a focus group meeting participants were asked to completed a brief survey collating demographic data about their courses and indicating their availability for a focus group meeting. The survey remained open from 5th August 2024–3rd September 2024. All participants were offered the opportunity to participate in the preparation of the manuscript as co-authors in a future publication of the study results, with the proviso they contribute to interpretation of the data, reviewing the manuscript, providing final approval of the manuscript and agree to be accountable for the work, in line with journal criteria.

## Procedures

An online survey was developed jointly by the lead and final author, AV and JK, to address research question 1; What support and training are currently being provided for pre-registration student SLTs on the topic of dementia? Participants were asked to complete the online survey hosted on the UCL approved online platform Qualtrics [39] answering 16 questions providing demographic information about their courses and the dementia teaching they provide (level of course- BSc,

MSc, apprenticeship, how many hours of dementia teaching, who does the teaching, what is included in the teaching and placement opportunities. They were also asked to provide a brief comment on the local challenges around dementia teaching and whether they had any relevant resources they felt were unique and useful to share (see supplementary S2 File for the full list of survey questions).

Survey participants who expressed an interest in attending a focus group by sharing their email address received email-invitations to one of two 90-minute focus meetings, held on the online videoconferencing platform Zoom. The email provided information on all organisational issues (i.e., time and date, calendar entry, videoconference link, anticipated duration, privacy and request for response). Researchers met prior to the meeting to test out technology on the day, introductions allowed for testing of microphones and for participants to settle into the meeting. Participants were able to see and talk to each other and the facilitator, who was one of the researchers, during the meeting. All participants reported they were familiar with zoom. The chat was also and was used to share the focus group questions in written form as well as verbally. At the end of the first focus group a debrief was held between AV, JK and RT to identify any issues with moderation.

Participant demographic data about individual focus group participants were not collected, and it was agreed that all quotes would be carefully anonymised to reduce the risk of identifying the views of individuals or institutions and to allow for open and honest conversations. Focus group meetings were facilitated by a speech and language therapy researcher (co-author RT) not currently involved in dementia education, but familiar with the clinical area. Researchers AV and JK met to discuss the results from the initial survey questions and develop the focus group topic guide. Survey respondents noted that dementia was a difficult subject for some students and not all universities reported teaching the same dementia related topics. Some universities reported that student SLTs were able to access clinical placements within the field of dementia, but others were not. While the survey addressed the first research question; summarising the support and training currently being provided for pre-registration student SLTs on the topic of dementia, it did not provide details of the experiences or opinions of those lecturers in delivering this training, and exploring what should be taught to improve confidence and competence in student SLTs to enable them to work with people with dementia. For example, the survey data did not explain the variation in placement experiences and whilst topics within the teaching were listed there was no depth in terms of the rationale for these choices, and why this was relevant in terms of improving the competence in student SLTs to enable them to work effectively with people with dementia. To provide in-depth, narrative-based explanations for the initial findings AV and JK drafted topic guide question to consider these areas, thereby addressing research questions 2 and 3. Questions probed reasons for the variation in placement opportunities, rationale around priorities for current teaching and what lecturers thought should be taught to improve confidence and competence with context and reasons (see supplementary file 3 and 4 in the supplementary materials for the topic guide and full breakdown of survey data). Notably answers to the final two questions from the survey (barriers and resources for sharing) were felt to be too brief to allow for interpretation. The focus group topic guide therefore aimed to invite participants to provide deeper, richer responses to enable fuller interpretation. The topic guide draft was shared with co-researcher RT, an experienced speech and language therapist for feedback and revised with his feedback re clarity and use of prompts. The final topic guide (see supplementary S3 File) comprised seven open questions, with several accompanying probes, and contributors were encouraged to expand or clarify responses where required by the facilitator, RT. Consequently, the first focus group was attended by six participants, the second by 12 participants. AV and JK attended both meetings to provide technical support. AV and JK were also participants in the first focus group. Whilst this is a power dynamics risk, this was managed by setting a clear ground rule with the moderator RT that after the initial introduction AV and JK only contributed once other group members had spoken in the first meeting, thereby ensuring everyone had a voice. In the second meeting it was made clear to participants that AV and JK would not speak during the interview component. In total the focus groups included 18 participants. Importantly, in terms of reflexivity, AV, JK and RT explicitly discussed and reflected on their biases as colleagues during the development of the topic guide, acknowledging that in their roles as both facilitators

and researchers they might influence the interactions. Both focus group meetings (totaling approximately 183 minutes) were video recorded and automatically transcribed using Zoom software. Co-author MG watched the video and manually edited the transcription to ensure accuracy, and to check that all names, places and any other identifying information was anonymized.

## Analysis

Closed responses to survey questions were analysed using descriptive statistics including means and range. Open responses were explored to inform the focus groups as described.

The research team, AV, RT, MG and JK analysed the transcribed focus group data using reflexive thematic analysis [40,41]. MG, a speech and language therapy researcher specialising in dementia from Germany, was invited to participate in the analysis process to deepen the interpretation. Being mindful of AV and JK's relationship with the participants (as colleagues and peers), it was felt that MG could provide an alternative view. In line with guidance from Braun and Clarke [42] the researchers were striving to ensure their reflexivity and maximise their subjectivity through thoughtful analysis, and the addition of MG was felt to support a more inductive data driven approach through unmotivated looking and analysis. Authors AV and JK also produced a video recording of initial themes and shared this with all 16 remaining participants (excluding themselves), nine of whom responded. This enabled the further interrogation and refinement of identified themes. Fig 1 provides a detailed account of the analysis process.

## Results

1. *What support and training are currently being provided for pre-registration student SLTs on the topic of dementia?*

### Survey responses

18 respondents from 16 universities across the UK completed the survey, (this represented two universities where two participants submitted identical responses under different participant names). Data represented universities delivering BSc courses only (6), MSc only (2) both BSc and MSc (5) and all three (BSc, MSc and Apprenticeship courses (3). Teaching on dementia within each course ranged from 4.5–20 hours, with a mean average of nine hours. Further details are captured in Table 1.

2. *What are the experiences of the lecturers in delivering this training?*

3. *What do lecturers think should be taught to improve confidence and competence in student SLTs to enable them to work with people with dementia?*

### Thematic analysis of focus groups

Eighteen participants from the same 16 universities represented in the survey participated in the focus group meetings. Six main themes were identified in the qualitative focus group data that described the experiences and opinions of lecturers involved in teaching student SLTs about dementia; 1. Dementia is a vast and therefore complex topic, 2. There are biases about dementia within and outside the profession, 3. Students bias towards dementia can be shifted through exposure, 4. Teaching could be enhanced by threading dementia through the curriculum, 5. There are several tensions in teaching on dementia: Possibilities versus clinical realities now and in the future, and 6. Dementia teaching must focus on person centeredness.

The following provides a detailed description of each theme, accompanied by relevant quotations from participants (using the letters A-S as identifiers) across the two focus groups:

1. Dementia is **a** vast and therefore complex topic

Transcriptions Shared: The transcripts created by MG were shared with the research team. These transcriptions were read by AV, JK and RT to re-familiarize themselves with the data prior to coding

Initial Coding: MG, AV, JK and RT coded the transcripts by hand independently, identifying interesting points and possible patterns in the data, making notes and reflections during this process. See examples of transcripts with initial coding:

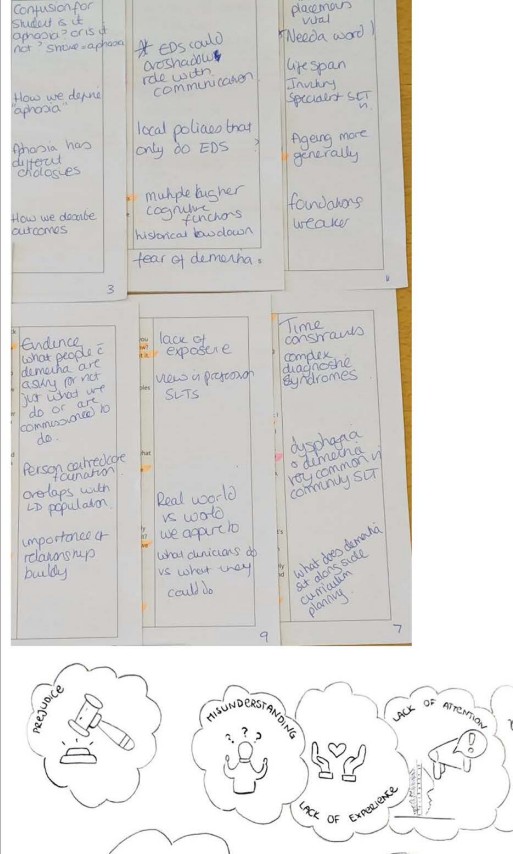

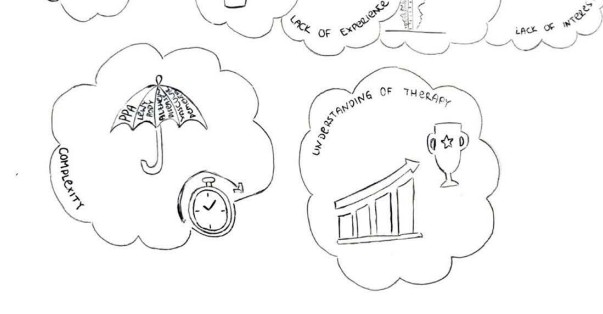

Example of researcher coding (NB: original transcript not shown to preserve anonymity)

Initial theme development: MG, AV, JK and RT met face to face to discuss their initial coding, thoughts and reflections of the data. Through this collaborative process initial themes were identified in the data, including how themes linked together. They produced a mind map to bring this together. AV and JK recorded a 10-minute video to explain the initial themes using

the mind map. This was shared with all participants (co-authors on this current study), and further reflections were requested.

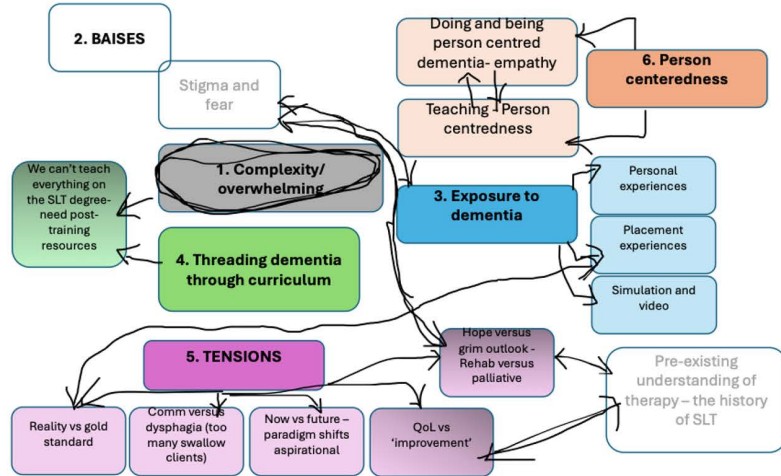

Refinement of themes: Having received reflections from participants MG, AV, JK and RT met again to discuss the reflections and themes and further refined the mind map and theme development. This focused on developing relationships between themes.

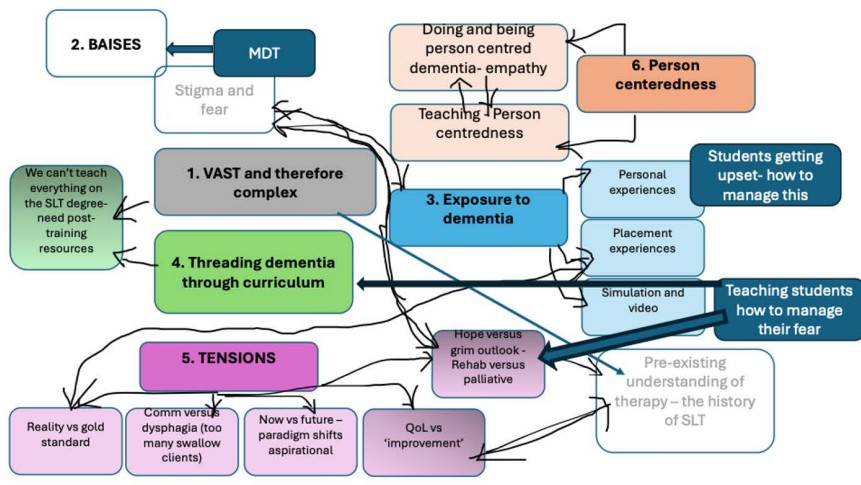

Finalising themes: AV refined the themes and produced a visual image to stimulate further discussion. At this stage AV, MG, JK and RT met again to agree themes names and corresponding quotes to illustrate themes.

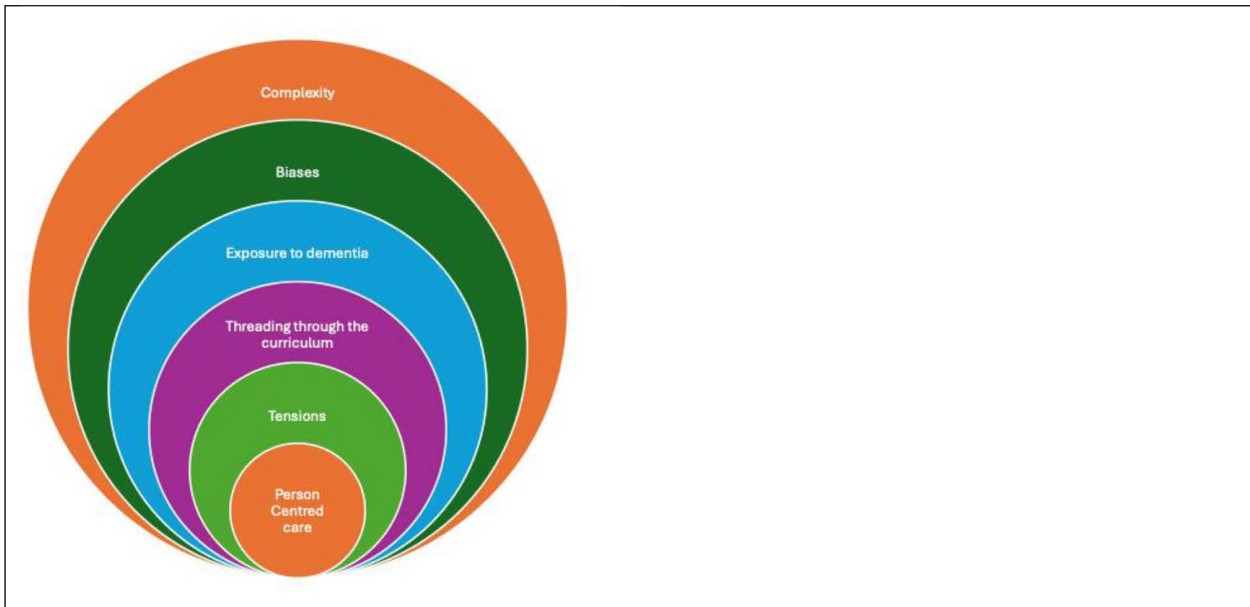

Writing of results: AV re-inspected the focus group transcripts to ensure themes accurately represented the data, and selected further quotes to fully illustrate the identified themes during the writing process. The results were shared with MG, JK and RT before finalising the manuscript.

**Fig 1. Detailed account of the reflexive analysis process.**

Participants described that just like many other communication difficulties, dementia can be complex, making it difficult to teach. The breadth and heterogeneity of dementia were important component of this, meaning that participants felt dementia was a vast and expansive field, encompassing many different and often nebulous diseases that take time to explain to student SLTs:

*We have to help students to understand just how many symptoms can present and how different it is between individuals. And I think many students' minds are blown. [It] really requires a good meaty chunk out of teaching time to explain that complexity.* (participant O)

Students were felt to benefit from frameworks to explain new concepts, yet dementia is not easily explained in this way. Participants described that in order to provide comprehensive information, dementia teaching needs to cover a range of speech, language, communication, cognitive and swallowing abilities and difficulties. The range of presentations, at different stages of the dementia, caused by a variety of underlying diseases or aetiologies that changed over time, could mean that information was potentially complicated and confusing for students, e.g., *I think it gets a little bit muddied in this sort of general umbrella term.* (participant A)

Albeit somewhat overwhelming, foundational knowledge can be delivered through university teaching of SLT students. However, participants advocated that further development of knowledge is required beyond what it is possible to provide

**Table 1. Survey responses describing dementia teaching on speech and language therapy courses at 16 Universities across the UK.**

| Type of degree | |
|---|---|
| BSc Only | 6 |
| MSc only | 2 |
| Both BSc and MSc | 5 |
| BSc, MSc and Apprenticeship | 3 |
| Hours of dementia teaching on BSc | 10.4 (range 4.5–20) |
| Hours of dementia teaching MSc course | 7.5 (range 6–12) |
| Hours of dementia teaching on apprenticeship courses | 7 (5-10) |
| Taught by (NB some courses are taught by multiple lecturers): | |
| Lecturer who is a specialist clinical academic SLT in dementia | 8 |
| Lecturer who does not specialise in dementia but is an SLT by background | 4 |
| Guest lecturer who is a specialist clinical SLT in dementia & Lecturer who doesn't specialise in dementia but is an SLT by background | 3 <br> 1 |
| Lecturer who is a specialist clinical academic SLT in dementia & Guest lecturer who is a specialist clinical SLT in dementia | |
| Do students attend specialist dementia placements | |
| Yes | 1 |
| Some but not all | 12 |
| No | 3 |

NB: In line with UK National Health Service speech and language therapy employment banding specialist speech and language therapist denotes an individual with clinical experience in a particular area such as dementia

in the brief hours that are available in the training. Experience in the workplace was also felt to be vital to achieving this post-registration skill development. Historically teaching was felt to be inadequate to prepare people for clinical practice:

*When I did my training, we literally had. I'm not joking. We had 10 minutes at the end of the aphasia course on head injury and dementia, or 15 maybe.* (participant E)

Developing skills in working with people with dementia who have communication needs was felt to be a new area within the profession, making it vulnerable to reductionist attitudes or being overlooked in preference for swallowing:

*It's still a relatively new area within our profession. There's a lack of tradition in our profession compared to say stroke and aphasia and the danger for the future is that... EDS [eating, drinking and swallowing] within dementia overshadows communication within our whole profession.* (participant R)

The newness of working on communication with people with dementia was considered to add to the complexity, resulting in it being a less comfortable and more challenging field of work:

*Maybe for some of us we feel a bit more reticent, because it's outside of our comfort zone, whereas the sort of traditional aphasia and stroke tends to be much more something that we're – we're confident with.* (participant A)

Participants attributed the feelings of discomfort to the historical nihilistic views that there is nothing that can be done for people with dementia. Whilst this view has changed, as research has demonstrated the positive impact of speech and

language interventions, this previous view was felt to be pervasive, with a continued impact on both the profession and the literature in this field:

*I think it's historically been quite underrepresented within the profession. So, I think you know, for me when I first started teaching, we didn't have very much on dementia, and I think you know we've had to kind of try and change perceptions around dementia. And obviously that's helped with it being quite high on the kind of political agenda as well..... I think there's a lot of fear of you know. What do we do with people who have these progressive conditions.* (participant J)

Participants conveyed a sense that the complexity in teaching communication in dementia did not only rest with the condition itself, the diagnostic and prognostic difficulties and the combination of language and cognitive communication difficulties, but with the historical attitudes and lack of research evidence about the role of speech and language therapy in dementia:

*People are not giving [it] attention. They're not calling it aphasia. It's much more new into the field, right? The post-stroke aphasia has been around from Second World War very well entrenched, but dementia [and] PPA is like 1990s, and in the last 20 years it has picked up so it will take time to for, you know, clinician academics to catch up.* (participant C)

2. There are biases **about** dementia **within** and **outside the** profession

Biased views about the role of speech and language therapy in dementia were felt to be genuine barriers to teaching students. Importantly, the historical attitudes towards dementia within the profession were only felt to be one element of this. This bias was greatly exacerbated by broader biases in society about dementia that presented negative representations of people with dementia in the media:

*The societal stigma around dementia as being a living death, and you know all of those terrible….I won't go into [charity] marketing campaign around dementia but I think there's a lot of stigma that that dementia is this really distressing? I say [to the students], living death is often how it's described in the media, and I think that's unhelpful and inaccurate. They should also say that people can have really positive lives with dementia, but I think that probably adds to the fear and upset that people have when they think about dementia.* (participant N)

The stigma of dementia was highlighted as a problem impacting attendance at lectures. Participants reported students frequently approached them prior to and during lectures to share concerns, anxiety and fears:

*I've had experience of many years of teaching speech and language therapists at all stages of their career about dementia, and there's always been tears. Before I'd ever heard the term trigger warning I started to introduce that because I found that it was grandparents. It was parents. It was people's own worry about their own memory problems.* (participant R)

Participants reported spending significant amounts of time trying to unpack these pre-existing biases in lectures. A participant from one university had consulted with people with lived experience of dementia who had emphasised working through these biases as a priority for teaching. Several participants described innovative ways of encouraging students to do this in lectures, using reflective tasks and comparing these to photos and AI representations to challenge students' assumptions:

*I wonder if now, with the use of AI, we can be utilizing that more effectively within our teaching as well people with dementia. So, for example, I, when we were doing the challenging kind of stereotypes of older people. I got AI to*

*generate a whole bunch of images of people with dementia, and we talked about how they were all wrinkly old people.* (participant D)

The concept of having to unteach biases included a feeling that most other teaching related to children and adult communication needs focused on restitution and rehabilitation in terms of the SLT role. This meant that when teaching dementia (as with other progressive conditions) there had to be work done to re-calibrate students' knowledge and expectations around intervention goals and therapy approaches:

*Looking at your outcome measures and what you're trying to achieve out of any therapy that you're providing, because with stroke, you are looking for some sort of restorative in some form or some resolution or improvement, whereas with dementia (…) you may be able to resolve some issues in the very short term but actually, you're looking at that longer term picture where the chances are you are going to have continued difficulties and increased difficulties throughout that time. (…). It's planning for that as well as kind of working on the here and now.* (participant E)

Some participant lecturers reported that it was challenging to make dementia an exciting subject area to teach because of the progressive trajectory of the disease. One participant explained that dementia could potentially be viewed as less exciting and held negative connotations for students:

*When we're teaching about stroke and aphasia, and we're talking about rehabilitation, and the students get quite excited about the potential for rehabilitation, and then we move on to dementia. I think that generally the understanding in dementia is that the picture is quite grim, and that people are progressing, and it doesn't seem as exciting, perhaps, to the - to the students as the potential for Rehab.* (participant A)

3. Students bias towards dementia can be shifted through exposure

Participants described a need to shift student's perspectives by providing them with exposure to dementia, thus creating new experiences and changing biases. Experiencing dementia through the media or news was not felt to provide any help nor equip the students to meet a person with dementia:

*I think it can address the fear dementia sounds really scary, and people, if you've only ever read a book about dementia, you could feel very ill equipped to handle an interaction with someone with dementia.* (participant R)

Unfortunately, exposure to dementia prior to the course in some instances reinforced stigma and distress. It was not uncommon for students to bring personal experiences related to dementia to teaching staff, such as upsetting family experiences. Whilst students were found to have personal experiences in relation to several communication disorders that are taught on speech and language therapy courses, personal experiences of dementia were particularly variable, at times distressing and often instilled inaccurate representations of the condition or the role of the SLT:

*A lot of students seem to have accurate or inaccurate fairly strong ideas about what dementia is, or strong personal stories or links...or a strong sense of what the SLT role might be in dementia.* (participant R)

Despite the general opinion that more experience would improve understanding of dementia, students reported inconsistent experiences of dementia on placements. These could be polarising and serve to either enhance the students' positive opinions of the role of the SLT in dementia or provide pessimistic and traumatising experiences as these contrasting statements illustrate:

*Exactly like you're saying they they'd say the most things like we didn't expect them to have a sense of humour. You know, some people with dementia are actually really fun. And they say that they're really fun to work with.* (participant F)

*I've seen in the acute setting with taking students round, and there was a lady who was accusing the student as soon as she walked in about stealing her fruit, you know so, and just being able to handle that and not sometimes [feel] that, that's a bit scary for them. You know. How do I feed into this reality. How do I behave?* (participant G)

*I think the first time I ever went into a care home was [as] a speech and language therapist, you know (…) I still remember it now of being, you know, quite an intense experience really and unfamiliar. And so, for a lot of students that might, that might be the first time they've been into a care home and potentially so that that could, that could be a challenging experience.* (participant S)

Some placement experiences can trigger memories of personal experiences and become interwoven with personal experiences:

*The emotional impact that some of our students have fed back to us. I think some students have fed back when they've had a relative who has had dementia and has perhaps passed away, and that experience of going and visiting them in the care home and things, and then having a similar experience on placement that has brought up, you know, certain feelings for them that they found quite upsetting, and that we've kind of had to support them through.* (participant Q)

The reality of some clinical services was felt to exacerbate stigma and fear amongst student SLTs, or even spread false beliefs about speech and language therapy not being at all effective:

*I've had some of the attitudes of the other speech language therapists quite problematic in terms of you know, like the kind of what's the point of it. Kind of oh, you know what is this? It's not proper work.* (participant F)

Even when they'd had positive experiences, some students perpetuated negative views of placements focusing on communication therapies as somehow inadequate because it is not 'direct' therapy with the person to improve their communication abilities. This was felt to be associated with stereotyped views of the SLTs role or prior training in impairment or restitutive interventions skewing students to describe approaches such as life-story work or communication partner training as 'indirect' work and by implication therefore not 'real' speech and language therapy:

*The proper work is [perceived as] direct work with the patient where people improve and then this other stuff (...) Even the term indirect boils my blood, because it's still direct with somebody, isn't it? (…) And there's this kind of you know, 'we couldn't do that. So we did this'. So it's already sort of prefacing it as something that's of a lower standard.* (participant C)

Students had reported to participants that placement supervisors would occasionally not allow them to see people with dementia as they were 'too complex'. Or placement supervisors themselves felt less confident in working with people with dementia on communication:

*They're de-skilled a bit with doing so much dysphagia now and such little communication, that actually, that that can be a bit scary, can't it?* (participant F)

However, with expanding cohorts of students, there were not enough placements available for all students to meet people with dementia. This was often attributed to misconceptions about placement purposes among SLTs in clinical practice

and participants felt responsibility for forging improved relationships with placement providers to assure them that these were valued learning opportunities for students:

*Not all mental health services are taking student placements. And I think that is a big problem for the profession. And I think that is something that we kind of need to think about, and that seems to be a pattern across the country. Some places will take students very regularly, other places won't, and this. I'm never quite sure whether it's the therapist, or whether they've got some misconceptions about what they think the universities expect of them, and our places do quite a lot of work. We've tried to support those discussions, and you know, so it. And then that therapist changes.* (participant F)

Given some students did not see anyone with dementia on clinical placement, alternative means of providing a range of experiences of dementia was felt to be necessary. Purpose-built opportunities for all students to meet people had been introduced at some universities with widespread positive effect, for example running groups on campus.

Other participants had explored opportunities through non-NHS providers, and non-SLT providers:

*We are actually trying to find other solutions. For example, working with charities, you know, placement with charity sectors as one option [such as] carer groups. So, it's not directly with the patient themselves, but actually with the carer and other support groups which supports dementia.* (participant C)

Simulation experiences and video recorded patient experiences were considered possible alternatives:

*So, a lot of the students will come to the end of their last placement close to graduations, that they might have not had the opportunity to see any cases with dementia. So, as part of the teaching, it's important to take this into consideration. Present some videos, some cases, just to make sure that they get that exposure.* (participant K)

Some participants also emphasised the importance of trying to inspire students, and enthuse them during their teaching, to dispel myths and emphasise the vast range of opportunities to develop the profession in a positive way:

*I'm often saying things like: Oh, you know, this is the biggest expanding caseload for speech and language therapists, because more and more people are getting dementia. We're going to see them all on our caseload. And you know, the research field is expanding as we speak. So, I try, and I talk about it that way, and there's so much opportunity.* (participant E)

4. Teaching could be enhanced by threading dementia through the curriculum.

Participants felt there was an opportunity to increase teaching on dementia by threading dementia teaching through the curriculum in a practical way to teach several important topics. In fact, dementia was seen as a useful vehicle to teach on topics such as neuroanatomy, dysphagia, decision-making and mental capacity, ethics, family support, cognitive communication disorders, and multidisciplinary working:

*I've been trying to influence some of my peers on other modules. So, for example, the stats teaching when the stats module coordinator asked for examples of different research methods. I found a paper on dementia, and I was like: Use this! Use this! And, for example, we're doing some more qualitative [teaching] again I found some qualitative research methods. For the mental capacity teaching I've tried, which I do part of I'm always trying to get examples of dementia clients into that. (…) I coordinate the research module in our course. So, I'm just. I often try and use dementia as an*

*example across in terms of showcasing the research (…) It's about threading, isn't it? Potentially, and influence across.* (participant E)

Dementia was considered a useful clinical example to support development of complex clinical skills as well as building resilience. By including dementia in several if not all areas of the curriculum some universities already reported successes in foregrounding dementia as a key clinical area for the profession:

*So the dementia theme kind of runs through the curriculum...It doesn't have its own module... we sort of drip feed a little bit more dementia in there as well, so that it's not seen as a separate thing to everything else. And it's a key clinical topic.* (participant B)

Threading dementia through the curriculum was felt to be a natural progression within the teaching curriculum. It was felt to benefit several other areas including dysphagia, Parkinson's disease, Motor Neurone Disease and other progressive disease groups:

*I touch on it in loads of other spaces as well. (…) the growing kind of eating, drinking, swallowing components that we're teaching on. Dementia comes up all the time. And yeah, you just find that you are talking about it in lots of different spaces.* (participant M)

From a practical perspective this was often achieved through case examples:
*We use examples of dementia cases, right? So that's other place where I felt we could embed the idea of dementia across the curriculum.* (participant C)

Given the rising number of people with dementia participants proposed the need for a radical change within the curriculum to ensure representation of a condition of growing importance to the profession:

5. There are several tensions in teaching on dementia: Possibilities versus clinical realities now and in the future

Throughout both focus groups participants expressed a tension in teaching dementia. Some of the concerns were about the lecturers' own confidence and skill set in teaching this topic:

*Some of us we feel a bit more reticent, because it's outside of our comfort zone, whereas the sort of traditional aphasia and stroke tends to be much more, something that we're – we're confident with.* (participant A)

Other participants highlighted tensions in where the teaching was placed within the curriculum, for example across and within progressive and non-progressive acquired conditions, emphasising that whilst there are commonalities across communication disorders, there are also significant differences, as expressed in this example where dementia teaching was grouped alongside the teaching of head injury:

*It got shared between myself and head injury and we found that when we were trying to collaborate to write this session together, we really couldn't, although we agreed on what is cognitive communication. How does cognition impact actually the kind of assessment was very, very different, and the purpose was very different.* (participant D)

There was felt to be a significant tension between the current clinical reality, where people with dementia are often deprioritised on clinical caseloads, compared to the future aforementioned "paradigm shift" (participant A), where people with dementia could access speech and language therapy for communication needs. This shift was felt to be both inevitable as our population ages yet remained currently aspirational. Despite this, participants felt strongly that there needed to be some change:

*I just wanted to highlight that with the increasing numbers of people living- living to an older age, there's quite a – there should be quite a paradigm shift in our education, because we are going to be seeing more of the people living with dementia, whether it's in community settings or the hospital. And I think there's quite a lot of work to be done with changing perceptions and altering perceptions and assumptions on people with dementia and their ability to express themselves and understand. (participant G)*

It was acknowledged that SLT provision for people with dementia in the UK was underfunded, particularly for communication services and so participant lecturers were faced with the dilemma of whether to teach the current clinical reality or what the evidence base indicated was possible given adequate resources. Awareness and teaching of both was acknowledged as important:

*What I teach in my sessions versus what is the reality for services that are very stretched and have students out on placement. Sometimes it's getting that balance right between what I'm saying can be done and what we can do as speech therapists and what we can achieve versus what is funded, commissioned. You know what we actually have time for, and that little bit of mismatch.*

*Gold standard versus sometimes the more realistic sort of situation. (participant M)*

It was acknowledged that communication was often deprioritised in the current reality, due to the overwhelming numbers of clients with difficulties with eating, drinking and swallowing difficulties. This was often reflected in the profession and the placements, where dementia placements focused on dysphagia, with no communication experiences:

*EDS [eating, drinking and swallowing] within dementia overshadows communication within our whole profession. (participant R)*

A lack of awareness of the role of the SLT within the care pathway was also an issue:

*I went to a care home to see a client with dementia and said I was there for*

*communication, and the staff at the home said, I didn't know you did that [they] thought, speech and language therapy was swallowing. (participant B)*

*People don't have awareness [of SLT] only in PPA or memory clinic they are, you know, they see the value. (participant C)*

Understanding the 'gold standard' for SLT practice was considered a method for empowering future generations of SLTs to advocate for, and plan for the right kind of services:

*I make a real point of saying; this is the ideal. This is what we want you to aspire to and actually want you to advocate for and you'll be in a position once you're in clinical practice to start trying to do that and putting in business cases. (participant D)*

6. Dementia teaching must focus on person centeredness

At the center of the discussion on quality of life was person centered care. This was an inarguable key component of all teaching on all communication difficulties, but felt to be exemplified within the field of dementia care:

*We're teaching person centered care and kind of holistic care in all different etiologies. But I think particularly in dementia. It's so important to think about person centeredness and talk to our students about person centeredness, and also, including the people around the person, with dementia as well. In our interventions. (participant H)*

                                                                                    

Person centered care was described as an aim of all teaching and care for people with dementia and therefore was considered a foundational approach within teaching. Exposure to dementia was identified as essential in developing the skills of empathy that are at the heart of this approach.

*Those sorts of experiences really resonate with them, and then their management and their assessment, and all of that is, is more likely to be influenced. I think, developing empathy rather than an academic understanding or pity from being able to relate to a person we can teach person centredness.* (participant R)

As such person centered care required staff to teach students both how to understand the theory and then apply it in their practice. This was considered the lynchpin and brought together teaching with placement experiences and exposure to dementia:

*We asked some of our experts by experience what we should cover in teaching. And so, their feedback was (...) about acknowledging stigma and challenging those stereotypes that people might have around dementia. The other thing that they highlight as well was how to how to talk to us with a capital "us", and rather than talking to family carers. (...) And then the other thing that they highlighted was about valuing them as individuals, and how it's important to see the value that people living with dementia can contribute. So, it's not just about focusing on disability but focusing on ability.* (participant N)

Despite this overwhelming agreement amongst participants that person centred care was a central component for teaching, one participant flagged that there is not a lot of literature that explains how this is operationalised in terms of delivery of clinical speech and language therapy. Participants acknowledged that it takes time and practice to develop person centred skills:

*It's a very practiced clinician that feels comfortable going into a setting and not having to whip out a form to fill in to guide them.* (participant B)

The concern being that SLTs often continue to assess, or take detailed case histories instead of spending time with families and getting to know the person deeply:

*Person centred care and how you know some of them get. And then some of them don't. And it's like, you know that kind of move away from the test battery, you know, and it and it's at what? At what point I mean it. And it's that underpinning stuff.* (participant F)

*It's not a case history taking. You're building a relationship with them. And you're seeing what their issues are from their perspective. But if I'm honest, (…) I don't know where to find the literature to help me with that.* (participant F)

However, in this discussion participants started to describe the clinical practice of being a person centered speech and language therapist as "building a relationship" (participant F) with the person and their family and getting to know people's individual communication difficulties, as expressed succinctly here:

*Drilling down into what are the unique communication needs or difficulties of someone with one of the dementias.* (participant A)

## Discussion

The survey data collected in this study demonstrates the range in hours, opportunities and content on the topic of dementia that student SLTs receive across UK universities. Focus group data provided qualitative context to help understand

this data through the experiences of lecturers, specifically providing some novel examples of good practice and common barriers, e.g., in accessing placements. Participants lecturers' identified an urgent need for a "paradigm shift" to ensure the profession is equipped with the appropriate skills to work with the increasing number of people affected by dementia. Participants advocated that additional time needed to be allocated within the curriculum to meet this growing need and role. This opinion aligns with current and anticipated prevalence data demonstrating the increase in the number of people living with dementia in the UK [2]. Most people living with dementia experience some kind of communication difficulty, whether it is speech, language or cognitive communication difficulties [43,44], and many will also experience swallowing difficulties [45]. Exploring current and projected incidence of disease could inform decisions about teaching content that best equips student SLTs into the future.

Whilst participants reported a range of time allocated to teaching student SLTs on dementia, ranging from 4.5–20 hours, they identified similar challenges. Participants related these challenges to a multitude of factors including the historical focus with speech and language therapy on restitution and 'improvement' trajectories when working with communication disorders. The speech and language therapy profession has worked with people with stroke aphasia since prior to the second world war, developing a large literature of research evidence on restitutive rehabilitation approaches, functional participation and activity focused interventions [17,19]. Misperceptions that people with dementia are unable to benefit from these interventions have perpetuated nihilistic attitudes that there is nothing that could be done [5,46–48]. Despite this the number of people with dementia referred to speech and language therapy have slowly increased [25,49]. In parallel several pieces of research recently undertaken have highlighted what people with dementia and their families feel is important to their lives: to maintain communication and conversation with family and friends [28,29]. It is thus perhaps unsurprising that participants in the current study felt strongly that the future of the profession should include delivering more interventions to address both swallowing and communication needs and enhance interactions and relationships for people affected by dementia.

The current clinical reality was identified in this study as one of the most significant barriers to upskilling student SLTs to work with people with dementia. By current clinical reality, we mean the lack of resources, both staff and time, and restrictive service criteria, that often exclude people with dementia from communication support, and impact negatively on clinical placement experiences for students. Similar to research studies in the USA [30] and Brazil [32] clinical experience with people with dementia was felt to be of most benefit to allow SLTs to develop competence and confidence in working with people with dementia. This aligned with the best practice principle for SLTs working with people with language led dementias "Getting to know people deeply" [50]. However, current clinical placements were not able to provide the required experiences. Participants in the study identified that this was attributable to the current clinical reality in the UK, which did not necessarily allow time to work on communication skills with people with dementia, as speech and language therapy services are often structured to prioritise management of swallowing difficulties. This had led participants to create several novel approaches to delivering opportunities for student SLTs to meet and work with people with dementia by setting up specialist university clinics, collaborating with charities locally, by inviting people with dementia into lectures, through video and virtual reality opportunities. Some of these contrast with the long-term experiences described by Mahendra et al [22] and Banerjee et al [33,51], who suggest that developing proficiency in working with people with dementia requires students to have opportunities over a longer period of time. This discussion emphasises a need for future research to examine how best to develop competence and confidence amongst student SLTs in working with people with dementia.

Participants in the study unanimously agreed that there is a need for a basic set of core components to inform current teaching on dementia for student SLTs in the UK. Foundational to this is the need to teach person centered care, and to describe what this looks like clinically in terms of getting to know people deeply, and using this to then select and deliver assessments and interventions built around individual need. To describe how to get to know people deeply, five core components or 'pillars' were synthesized during the manuscript writing by AV, with feedback from

**Person centered care for speech-language therapy teaching on dementia**

1. Challenge stereotypes around dementia

2. Focus on speech, language and communication across dementias

3. Teach them to build a relationship with people affected with dementia

4. Teach goal setting for a progressive trajectory

5. Prepare them to advocate for gold standard

**Fig 2. Five core components – the pillars for teaching student speech and language therapists to work with people with dementia have been synthesized from focus group discussions.**

JK, RT and MG and further refined with feedback from remaining co-authors (see Fig 2). The participating lecturers in this study have formed a network of speech and language therapy dementia lecturers and agreed to meet twice annually to continue to advocate further to develop teaching on the subject. The group agreed that ideally dementia should be woven throughout the curriculum and across modules, where possible because the condition is a vehicle for a range of transferable skills. Future research should work with people with dementia, to further refine the core components for teaching on this subject for student speech and language therapists. The five core components comprise:

1. *Challenge stereotypes around dementia*: Spend time challenging stereotypes and addressing the stigma of dementia FIRST when teaching students about dementia. This might include reflecting on student's assumptions about the appearance, age and communication skills of people with dementia, and their ability to participate in therapy.

2. *Focus on speech, language and communication across dementias:* Students require a foundational knowledge of the range of conditions that fall under this umbrella term, with a focus on speech, language and communication across dementias. This should include all the main dementias as well as rarer types, and should provide key information about speech, language and communication profiles and how these may interact with other key symptoms.

3. *Teach them to build a relationship with people affected by dementia*: Students need to be taught to spend time building a relationship with people affected with dementia, this means getting to know them, before reaching for formal assessments. This is best taught through experiences of meeting people with dementia and their families within the community, finding out about their past and current lives (this may be achieved through placements, co-teaching, and video samples)

4. *Teach goal setting for a progressive trajectory:* Students will need to be taught how to set goals that take into account a progressive disease trajectory, and should include activity, participation and environmental interventions. This may mean outcomes do not focus on improvement, but on maintenance, strategy use, quality of life and participation, or the strategies of the people in their environment.

5. *Prepare them to advocate for gold standard*: It is important to teach student SLTs about their role in supporting people with dementia in terms of communication *and* eating, drinking and swallowing difficulties. This will ensure they are prepared for the current clinical reality, but also able to meet the needs of current and future people affected by dementia and enable them to advocate for gold standard services. This may include teaching them about using a range of evidence sources, service user involvement in service design and outcomes that are important to a range of key stakeholders.

## Strengths and limitations

Work undertaken in the research study has brought together speech and language therapy lecturers from across the UK to explore teaching to ensure students develop competencies that meet the needs of people with dementia and their families. Two focus groups were held, with a much larger group of participants attending the second meeting, meaning it is possible that not all participants were given equal opportunities to respond. Whilst 18 representatives from 16 universities were involved, it was not possible to engage all 25 higher education institutions in the UK and therefore this may be a limitation of this study. To protect anonymity, demographic data collected about focus group participants was presented in a way that prevented cross referencing. This may have reduced the possibility of developing more in-depth insights to participants individualized opinions and linguistic, cultural or ethnic biases. Despite the action taken to address the power dynamic, the presence of the first and last author at the focus group discussions may have influenced what participants did or didn't share. Importantly, however, almost all focus group participants were able to contribute the time and work to be co-authors on the paper, demonstrating their commitment to better address the needs of people affected by dementia in student speech and language therapy training. Whilst it was not probed or raised during the focus groups, future research on teaching student SLTs must explore linguistic, cultural and ethnicity needs of people affected by dementia. No people affected by dementia themselves were involved in this study, so it is difficult to know if the themes and components identified will really meet the needs of people with dementia. Future research should explore this more fully.

## Conclusion

People affected by dementia are one of the biggest expanding caseloads for SLTs, highlighting a need for a paradigm shift in teaching student SLTs. Clinical experiences were identified as one of the best methods to support students to shift their perspectives and develop confidence in working with people with dementia. Lecturers training student SLTs identified that teaching must make provision for the current clinical reality and an aspirational gold standard. Person centered care has been identified as foundational to all teaching on the subject. To ensure that student SLTs can truly meet the needs of people affected by dementia future research must be undertaken jointly with people with lived experience.

## Supporting information

**S1 File. MMAT tool annotated to demonstrate how manuscript follows the guidance.**
(DOCX)

**S2 File. Survey questions.**
(PDF)

**S3 File. Focus group topic guide and detailed overview of the teaching content reported by respondents.**
(DOCX)

## Acknowledgments

We would like to acknowledge those participants who responded to the study but were unable to attend the focus group meetings. We would also like to thank all participants in the study, 17 of 18 of whom are co-authors on this study, and lecturers at universities teaching student SLTs on working with people with dementia. The remaining two participants were positive about the components, and we are extremely grateful for the time they contributed to focus group and email discussions.

## Author contributions

**Conceptualization:** A. Volkmer, Jackie Kindell.

**Data curation:** A. Volkmer.

**Formal analysis:** A. Volkmer, Reem S.W. Alyahya, Hannah P Atkinson, Arpita Bose, Hannah Britton, Lindsey Collins, Katie Earing, James Farraday, Dharinee Hansjee, Mary Heritage, Sophie Mackenzie, Sally Pratten, Catherine Tattersall, Daniel Underdown, Allison Virgilio, Nicola Withford-Eaton, Mirjam Gauch, Richard Talbot, Jackie Kindell.

**Investigation:** A. Volkmer, Jackie Kindell.

**Methodology:** A. Volkmer, Jackie Kindell.

**Project administration:** A. Volkmer.

**Resources:** A. Volkmer.

**Validation:** A. Volkmer, Reem S.W. Alyahya, Hannah P Atkinson, Arpita Bose, Hannah Britton, Lindsey Collins, Katie Earing, James Farraday, Dharinee Hansjee, Mary Heritage, Sophie Mackenzie, Sally Pratten, Catherine Tattersall, Daniel Underdown, Allison Virgilio, Nicola Withford-Eaton, Mirjam Gauch, Richard Talbot, Jackie Kindell.

**Writing – original draft:** Katie Earing, Allison Virgilio, Jackie Kindell.

**Writing – review & editing:** A. Volkmer, Reem S.W. Alyahya, Hannah P Atkinson, Arpita Bose, Hannah Britton, Lindsey Collins, James Farraday, Dharinee Hansjee, Mary Heritage, Sophie Mackenzie, Sally Pratten, Catherine Tattersall, Daniel Underdown, Nicola Withford-Eaton, Mirjam Gauch, Richard Talbot, Jackie Kindell.

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
