## [Decision Letter · Decision Letter 0]

3 Sep 2025

Dear Dr. Volkmer,

Thank you for submitting your manuscript to PLOS ONE. After careful consideration, we feel that it has merit but does not fully meet PLOS ONE’s publication criteria as it currently stands. Therefore, we invite you to submit a revised version of the manuscript that addresses the points raised during the review process.

We look forward to receiving your revised manuscript.

Kind regards,

Rohit Ravi, Ph.D.

Academic Editor

PLOS ONE

**Journal Requirements:**

1. When submitting your revision, we need you to address these additional requirements. Please ensure that your manuscript meets PLOS ONE's style requirements, including those for file naming. The PLOS ONE style templates can be found at https://journals.plos.org/plosone/s/file?id=wjVg/PLOSOne_formatting_sample_main_body.pdf and https://journals.plos.org/plosone/s/file?id=ba62/PLOSOne_formatting_sample_title_authors_affiliations.pdf 2. In the online submission form you indicate that your data is not available for proprietary reasons and have provided a contact point for accessing this data. Please note that your current contact point is a co-author on this manuscript. According to our Data Policy, the contact point must not be an author on the manuscript and must be an institutional contact, ideally not an individual. Please revise your data statement to a non-author institutional point of contact, such as a data access or ethics committee, and send this to us via return email. Please also include contact information for the third party organization, and please include the full citation of where the data can be found. 3. We note that this data set consists of interview transcripts. Can you please confirm that all participants gave consent for interview transcript to be published? If they DID provide consent for these transcripts to be published, please also confirm that the transcripts do not contain any potentially identifying information (or let us know if the participants consented to having their personal details published and made publicly available). We consider the following details to be identifying information:- Names, nicknames, and initials- Age more specific than round numbers- GPS coordinates, physical addresses, IP addresses, email addresses- Information in small sample sizes (e.g. 40 students from X class in X year at X university)- Specific dates (e.g. visit dates, interview dates)- ID numbers Or, if the participants DID NOT provide consent for these transcripts to be published:- Provide a de-identified version of the data or excerpts of interview responses- Provide information regarding how these transcripts can be accessed by researchers who meet the criteria for access to confidential data, including:a) the grounds for restrictionb) the name of the ethics committee, Institutional Review Board, or third-party organization that is imposing sharing restrictions on the datac) a non-author, institutional point of contact that is able to field data access queries, in the interest of maintaining long-term data accessibility.d) Any relevant data set names, URLs, DOIs, etc. that an independent researcher would need in order to request your minimal data set. For further information on sharing data that contains sensitive participant information, please see: https://journals.plos.org/plosone/s/data-availability#loc-human-research-participant-data-and-other-sensitive-data If there are ethical, legal, or third-party restrictions upon your dataset, you must provide all of the following details (https://journals.plos.org/plosone/s/data-availability#loc-acceptable-data-access-restrictions):a) A complete description of the datasetb) The nature of the restrictions upon the data (ethical, legal, or owned by a third party) and the reasoning behind themc) The full name of the body imposing the restrictions upon your dataset (ethics committee, institution, data access committee, etc)d) If the data are owned by a third party, confirmation of whether the authors received any special privileges in accessing the data that other researchers would not havee) Direct, non-author contact information (preferably email) for the body imposing the restrictions upon the data, to which data access requests can be sent 4. Please include captions for your Supporting Information files at the end of your manuscript, and update any in-text citations to match accordingly. Please see our Supporting Information guidelines for more information: http://journals.plos.org/plosone/s/supporting-information. 5. If the reviewer comments include a recommendation to cite specific previously published works, please review and evaluate these publications to determine whether they are relevant and should be cited. There is no requirement to cite these works unless the editor has indicated otherwise. 

Reviewers' comments:

Reviewer's Responses to Questions

**Comments to the Author**

1. Is the manuscript technically sound, and do the data support the conclusions?

Reviewer #1: Yes

Reviewer #2: Partly

2. Has the statistical analysis been performed appropriately and rigorously?

Reviewer #1: Yes

Reviewer #2: No

3. Have the authors made all data underlying the findings in their manuscript fully available?

Reviewer #1: Yes

Reviewer #2: No

4. Is the manuscript presented in an intelligible fashion and written in standard English?

Reviewer #1: Yes

Reviewer #2: Yes

**Reviewer #1:**  The manuscript addresses an important and timely topic—Dementia—and explores the experiences and perspectives of lecturers teaching pre-registration speech and language therapy students across UK universities.

It lays a solid foundation by outlining the existing, albeit limited, research on this subject globally, and clearly articulates the need for the current study. The inclusion of direct excerpts from interview responses significantly enhances the manuscript, providing readers with valuable insights into the lecturers’ viewpoints and experiences.

The findings of the study highlight five key principles that can guide the integration and enhancement of speech and language therapy (SLT) education in the area of dementia.

I have only a minor formatting suggestion: Several statements appear in bold throughout the manuscript, but it is unclear whether this formatting is intentional for emphasis or an oversight. This inconsistency may affect the overall readability. Additionally, there are a few spacing issues that should be corrected.

Overall, the manuscript is very well written. I am impressed by the authors’ attention to detail across all aspects of the study.

**Reviewer #2: ** I thank the editor for giving me this opportunity to review the manuscript titled, “Dementia is our 'biggest expanding caseload': Core learning for student speech and language therapists”. While the topic is of interest and relevant to the current context where SLPs are seeing an exponential increase in the number of dementia patients in their practice, the manuscript has several methodological concerns that need to be addressed. I have first given my general/overall concerns pertaining to the manuscript, followed by section-wise comments. I hope these comments will help improve the quality of the manuscript

Overall Comments

Study Design: The Authors have mentioned the study design as “explanatory sequential mixed methods”, which is reasonable, but I don’t see how the quantitative data is explained by the qualitative information here, which is the crux of a mixed methods study. Additionally, I don’t see a clear description of how survey findings were used to inform the topic guide and later develop themes. Describe where and how integration occurred at the analysis or interpretation stage. Map each research question to the phase of the study that answers it

Inconsistencies in quantitative data reported: Authors state two focus groups with 6 and 12 attendees (total 18), yet the Results header says, “Nineteen participants”. Make these numbers consistent throughout. In the survey, you report 16 universities responded, but in Table 1, the line “Both BSc and MSc” lists 8, whereas earlier text says 5. Check all these values and report them correctly

Procedure:

1) Briefly detail the Zoom setup. There is established guidance on conducting rigorous online focus groups that you can cite.

2) Transcription and verification. Indicate transcription approach, de-identification, and accuracy checks. It is mentioned in the appendix that it was done manually; however, it is nowhere mentioned in the manuscript.

3) Your write-up uses RTA, but some phrasing implies a goal to “maximize objectivity” and “triangulate themes” via participant feedback. In RTA, researcher subjectivity is a resource, not a problem to eliminate; “inter-rater reliability” or “theme validation” by participants is not required and can conflict with RTA’s epistemology. If you shared themes with participants, frame this as a stakeholder consultation that informed interpretive refinement, not as verification. Also, avoid presenting multiple coders as a validity check; instead, explain how analytic conversations among AV, RT, MG, and JK deepened interpretation. Update language accordingly. Data Triangulation essentially used data from multiple sources, not multiple assessors. This needs to be revised.

Potential Bias in data: V and JK attended both groups mentioned as tech support and were also participants in Group 1; this is a power-dynamics risk and may shape discourse. Describe how this was managed (e.g., moderator control, ground rules, whether AV/JK refrained from speaking during interviews).

Results: The results are exhaustive. Almost, I feel the results can be better represented using tables showing codes, themes, and sub-themes rather than a 12-page summary of the results. Quotations should not overwhelm your narrative. The bulk of your results section should be your interpretation and thematic synthesis, with quotations serving as evidence.

Section-wise Comments

Abstract: The research questions can be removed from the abstract, and the authors can briefly mention the aims and objectives of the study. A lot of points in the abstract six themes labelled 1.,2., 3., … and then five principles…1,2,3…

Introduction:

Line 82: The study begins with mentioning the marked increase in low- and middle-income countries. However, this point does not align with the research question. This can be mentioned later as a global concern, and first, the data pertaining to the area of study needs to be highlighted.

Line 102: The font style/ size seems to be different. Kindly cross-check and maintain consistency.

Line 106: Is this expansion into different fields and client groups the author’s perception? If not. Some relevant articles should be cited here to support the claim of this evolution of the profession across different fields.

Line 111: Again, “progressive neurological conditions….. have been a later addition” is a claim that needs to be supported with evidence from the literature

Line 117: Abbreviations appear without proper reference. In line 113, use the abbreviation RCSLT in brackets first. Similarly, NICE is to be expanded here.

Line 132: “It seems logical…” Instead of saying this, the authors should write from relevant literature regarding barriers and discuss briefly how teaching efficient communication can overcome these barriers

Lines 170 to 174: These look like research questions and not aims. Kindly rephrase or mention that the study aimed to answer the following research questions.

Method:

Line 176: Explanatory sequential mixed-methods study: An explanatory sequential mixed-methods design is a reasonable choice (survey → focus groups). However, the manuscript needs to demonstrate integration, not only sequence. Please make explicit how survey results built the focus-group guide (name the specific survey items that led to specific probes), and where and how quantitative and qualitative findings were integrated in interpretation

You cite the MMAT; please report how MMAT criteria were applied to judge quality across the two components (not just that you “followed guidance”). Even a brief appendix showing item-by-item judgments will help.

Line 190: An email was circulated to the Committee of Representatives of Education in Speech and Language Therapy:” Are these Email IDs available on the RCSLT website? If so, specify that; otherwise, mention how these email IDs were obtained. Example if obtained from official websites of HEIs or elsewhere

Line 193-194; Some explanation is needed here. What was the duration from sending out the emails and getting the responses? After how many days/ weeks did the authors mail the remaining six HEIs? Were any reminders sent, or did everyone respond to the first email? The timelines need to be specified. Response rate/ completion rate of the survey needs to be explicitly reported

Line 197: 24 from 19 HEIs.. Further distribution is needed to specify bias in the data. What was the distribution of these participants across 19 institutes?

Line 199: First mention that the study had different phases… a brief survey followed by FGDs, to explain the procedure.

Line 202-203: The authorship should be based on ICMJE guidelines. Please specify the roles and responsibilities of each author as per these guidelines. Giving out authorship based on participation is not only unethical but also risks potential bias in the study.

Line 205: “Senior” in terms of age or experience in the dementia related field. Please specify

Line 207-212: The questions are vaguely described here in the method section. Did the questionnaire have any sub-sections? The manuscript lacks information on the development of the survey. How were these questions identified, and who validated the survey questionnaire? Please report instrument development (content mapping to aims), validations, and any revisions or CVI scores.

Line 210: They were also asked to provide a brief comment on the local challenges around dementia teaching and whether they had any relevant resources they felt were unique and useful to share. Where are the findings from these sections reported? I suppose this would constitute qualitative information reported by the participants. How was this taken up for analysis?

Line 218: Who were these speech and language therapy researchers (RTs)?

Line 231: The first FGD was attended by 6 participants, and the second by 12. What happened to one participant (12+6=18 out of 19)?

Line 233: Though authors have reported the bias as being the participants, out of the six participants in the first FGD, 2 were the study authors, which could have significantly biased the obtained data in the first round. Hence, I am skeptical about how the conversations were steered in this FGD. Describe clearly concrete reflexive safeguards (e.g., did AV and JK refrain from speaking?). Also, the description of the participants and their representativeness from HEIs is lacking.

Line 244: Kindly specify how the transcription was done. Though it is mentioned briefly in the appendix authors should clearly describe the method here. Why was no software like Atlas TI used?

Line 245: Though the use of RTA is apt, the authors should

Analysis: The Data Triangulation Process looks more like reliability. Again, this is mentioned in the appendix; however, a description of how it was done is warranted in the methodology section

Results:

Line 261: Further distribution of how the 19 participants were distributed across the 16 universities needs to be specified.

Table 1: Incorrect descriptive statistics reported. Means are reported with a range. What is the meaning of “specialist”? As the readers may be from different backgrounds, the authors need to specify what they mean by this.

Line 271: Refer to the comment given for line 231

Line 272: Authors need to specify how these themes were identified. A general process in qualitative research involves the generation of initial codes, which are further divided into themes. The codes are nowhere mentioned in the manuscript (except in the appendix), so as a reader, I am unable to understand how these themes emerged.

Line 282: The following provides a detailed description of each theme, accompanied by relevant quotations from participants (using the letters A-S as identifiers): All paerici

The result section is approximately 24 pages. exhaustive and is not summarised in a way that qualitative research should be reported. Authors should identify

Discussion:

The authors can integrate the findings from the mixed method design in this section.

Line 783: The prevalence data reported is repetitive and already mentioned in the introduction. Kindly rephrase and avoid repetitiveness.

Line 836: Five core components or ‘pillars’ were synthesized from participants’ focus-group discussions. How was this synthesis done? There is no mention of such themes or codes in the analysis process. How did authors come up with this during the writing process of the manuscript?

Limitations of the study to be discussed in detail

Table and figures

Tables and figures need to be significantly revised. Table 2 is not referenced anywhere in the manuscript.

Appendices

Appendices should support information from the manuscript. Currently, the important information that is significant in understanding the analysis and interpretation of findings is missing from the main body of the manuscript and added in appendices.

**Do you want your identity to be public for this peer review?** For information about this choice, including consent withdrawal, please see our Privacy Policy

Reviewer #1: No

Reviewer #2: No

---

## [Author Response · Author response to Decision Letter 1]

20 Oct 2025

Many thanks to the journal editor and reviewers for consideration of our manuscript. We hope the following addresses any queries or concerns.

Journal Requirements:

We have addressed the additional style requirements. Many thanks for flagging.

2. In the online submission form you indicate that your data is not available for proprietary reasons and have provided a contact point for accessing this data. Please note that your current contact point is a co-author on this manuscript. According to our Data Policy, the contact point must not be an author on the manuscript and must be an institutional contact, ideally not an individual. Please revise your data statement to a non-author institutional point of contact, such as a data access or ethics committee, and send this to us via return email. Please also include contact information for the third party organization, and please include the full citation of where the data can be found. The data availability statement has been revised as follows:

The datasets generated and analysed during the current study are not publicly available in their entirety due to the terms of our ethical approval. Extracts can be made available on request by emailing the ethics chair at UCL Department of Language and Cognition on fiona.kyle@ucl.ac.uk

3. We note that this data set consists of interview transcripts. Can you please confirm that all participants gave consent for interview transcript to be published?

If they DID provide consent for these transcripts to be published, please also confirm that the transcripts do not contain any potentially identifying information (or let us know if the participants consented to having their personal details published and made publicly available). We consider the following details to be identifying information:

- Names, nicknames, and initials

- Age more specific than round numbers

- GPS coordinates, physical addresses, IP addresses, email addresses

- Information in small sample sizes (e.g. 40 students from X class in X year at X university)

- Specific dates (e.g. visit dates, interview dates)

- ID numbers

Or, if the participants DID NOT provide consent for these transcripts to be published:

- Provide a de-identified version of the data or excerpts of interview responses

- Provide information regarding how these transcripts can be accessed by researchers who meet the criteria for access to confidential data, including:

a) the grounds for restriction

b) the name of the ethics committee, Institutional Review Board, or third-party organization that is imposing sharing restrictions on the data

c) a non-author, institutional point of contact that is able to field data access queries, in the interest of maintaining long-term data accessibility.

d) Any relevant data set names, URLs, DOIs, etc. that an independent researcher would need in order to request your minimal data set.

For further information on sharing data that contains sensitive participant information, please see: https://journals.plos.org/plosone/s/data-availability#loc-human-research-participant-data-and-other-sensitive-data

If there are ethical, legal, or third-party restrictions upon your dataset, you must provide all of the following details (https://journals.plos.org/plosone/s/data-availability#loc-acceptable-data-access-restrictions):

a) A complete description of the dataset

b) The nature of the restrictions upon the data (ethical, legal, or owned by a third party) and the reasoning behind them

c) The full name of the body imposing the restrictions upon your dataset (ethics committee, institution, data access committee, etc)

d) If the data are owned by a third party, confirmation of whether the authors received any special privileges in accessing the data that other researchers would not have

e) Direct, non-author contact information (preferably email) for the body imposing the restrictions upon the data, to which data access requests can be sent

We can confirm that participants gave consent for deidentified excerpts from the transcripts to be published. We have amended the data sharing statement as follows:

The datasets generated and analysed during the current study are not publicly available in their entirety due to the terms of our ethical approval. Extracts can be made available on request by emailing the ethics chair at UCL Department of Language and Cognition on fiona.kyle@ucl.ac.uk

Amended

Thank you

Reviewer #1:

The manuscript addresses an important and timely topic—Dementia—and explores the experiences and perspectives of lecturers teaching pre-registration speech and language therapy students across UK universities.

It lays a solid foundation by outlining the existing, albeit limited, research on this subject globally, and clearly articulates the need for the current study. The inclusion of direct excerpts from interview responses significantly enhances the manuscript, providing readers with valuable insights into the lecturers’ viewpoints and experiences.

The findings of the study highlight five key principles that can guide the integration and enhancement of speech and language therapy (SLT) education in the area of dementia.

Many thanks for this positive feedback, it is useful to know that the work is considered timely and important by the reviewers.

I have only a minor formatting suggestion: Several statements appear in bold throughout the manuscript, but it is unclear whether this formatting is intentional for emphasis or an oversight. This inconsistency may affect the overall readability. Additionally, there are a few spacing issues that should be corrected. We have rigorously checked the manuscript but other than the use of bolding for headings e.g. methods, results, discussion we cannot see any bolding. We have used spaces and italic to show when text is a quote. I wonder if there has been a glitch in the formatting in the process of transferring it to reviewers. Sorry for not being able to be more specific.

Overall, the manuscript is very well written. I am impressed by the authors’ attention to detail across all aspects of the study.

We have aimed to be extremely rigorous and therefore really value this feedback. Thank you.

Reviewer 2.

I thank the editor for giving me this opportunity to review the manuscript titled, “Dementia is our 'biggest expanding caseload': Core learning for student speech and language therapists”. While the topic is of interest and relevant to the current context where SLPs are seeing an exponential increase in the number of dementia patients in their practice, the manuscript has several methodological concerns that need to be addressed. I have first given my general/overall concerns pertaining to the manuscript, followed by section-wise comments. I hope these comments will help improve the quality of the manuscript

Study Design: The Authors have mentioned the study design as “explanatory sequential mixed methods”, which is reasonable, but I don’t see how the quantitative data is explained by the qualitative information here, which is the crux of a mixed methods study. Additionally, I don’t see a clear description of how survey findings were used to inform the topic guide and later develop themes. Describe where and how integration occurred at the analysis or interpretation stage. Map each research question to the phase of the study that answers it

The methods section of the manuscript has been refined to explain in more detail how the survey was used to develop the topic guide:

Whilst the survey addressed the first research question; summarising the support and training currently being provided for pre-registration student SLTs on the topic of dementia, it did not provide details of the experiences or opinions of those lecturers in delivering this training, and exploring what should be taught to improve confidence and competence in student SLTs to enable them to work with people with dementia. To provide in-depth, narrative-based explanations for the initial findings AV and JK drafted topic guide question to consider these areas in more depth. Questions focused on reasons for the variation in placement opportunities, rationale around priorities for current teaching and what lecturers thought should be taught to improve confidence and competence with context and reasons (see appendices 2 in the supplementary materials for full breakdown of survey data and topic guide).The topic guide draft was shared with co-researcher RT, an experienced speech and language therapist for feedback and revised with his feedback re clarity and use of prompts. The final topic guide comprised seven open questions”

The beginning of the discussion has been refined to demonstrate integration of interpretation of analysis:

“The survey data demonstrated collected in this study demonstrates the range in hours, opportunities and content on the topic of dementia that student SLTs receive across UK universities. Focus group data helped provide qualitative context to help understand this data through the experiences of lecturers, specifically providing some novel examples of good practice and common barriers, e.g. in accessing placements. Participant lecturers’ identified an urgent need for a “paradigm shift” to ensure the profession is equipped with the appropriate skills to work with the increasing number of people affected by dementia. Participants advocated that additional time needed to be allocated within the curriculum to meet this growing need and role.”

We have also clarified and numbered and mapped each research question to each phase of the study:

1. What support and training are currently being provided for pre-registration student SLTs on the topic of dementia?

2. What are the experiences of the lecturers in delivering this training?

3. What do lecturers think should be taught to improve confidence and competence in student SLTs to enable them to work with people with dementia?

Inconsistencies in quantitative data reported: Authors state two focus groups with 6 and 12 attendees (total 18), yet the Results header says, “Nineteen participants”. Make these numbers consistent throughout. In the survey, you report 16 universities responded, but in Table 1, the line “Both BSc and MSc” lists 8, whereas earlier text says 5. Check all these values and report them correctly

Many thanks for identifying these errors, we have corrected the manuscript to reflect throughout that there are 18 participants in the focus groups and corrected the table to reflect that there were 5 universities with BSc and MSC courses.

Procedure:

1) Briefly detail the Zoom setup. There is established guidance on conducting rigorous online focus groups that you can cite.

A reference to the guidance has been included at the start of the methods and further details added to the methods regarding the zoom setup:

“The online focus groups are described in line with online focus group reporting guidance [ref]. “

“Survey participants who expressed an interest in attending a focus group by sharing their email address received email-invitations to one of two 90-minute focus meetings, held on the online videoconferencing platform Zoom. The email provided information on all organisational issues (i.e., time and date, calendar entry, videoconference link, anticipated duration, privacy and request for response). Participants were able to see and talk to each other and the moderator, who was one of the researchers, during the meeting. The chat was also and was used to share the focus group questions in written form as well as verbally. “

2) Transcription and verification. Indicate transcription approach, de-identification, and accuracy checks. It is mentioned in the appendix that it was done manually; however, it is nowhere mentioned in the manuscript.

Both focus group meetings were video recorded and automatically transcribed using Zoom software. Co-author MG watched and the manually edited the transcription for spoken errors, ensuring all names, places and any other identifying information was anonymized.

3) Your write-up uses RTA, but some phrasing implies a goal to “maximize objectivity” and “triangulate themes” via participant feedback. In RTA, researcher subjectivity is a resource, not a problem to eliminate; “inter-rater reliability” or “theme validation” by participants is not required and can conflict with RTA’s epistemology. If you shared themes with participants, frame this as a stakeholder consultation that informed interpretive refinement, not as verification. Also, avoid presenting multiple coders as a validity check; instead, explain how analytic conversations among AV, RT, MG, and JK deepened interpretation. Update language accordingly. Data Triangulation essentially used data from multiple sources, not multiple assessors. This needs to be revised.

Thank you for this suggestions. We have refined the description of the analysis in the methods section of the manuscript -

“MG, a speech and language therapy researcher specialising in dementia from Germany, was invited to participate in the analysis process to deepen the interpretation.”

“Authors AV and JK also produced a video recording of initial themes and shared this with all 16 remaining participants (excluding themselves), nine of whom responded. This enabled the further interrogation and refinement of identified themes.”

We have also taken this opportunity to move the detailed description of the thematic analysis to the main manuscript – now table 1.

Potential Bias in data: AV and JK attended both groups mentioned as tech support and were also participants in Group 1; this is a power-dynamics risk and may shape discourse. Describe how this was managed (e.g., moderator control, ground rules, whether AV/JK refrained from speaking during interviews).

The following has been added to the methods to explain how this was managed:

“AV and JK were also participants in the first focus group. Whilst this is a power dynamics risk, this was managed by setting a clear ground rule with the moderator RT that after the initial introduction AV and JK only contributed once other group members had spoken in the first meeting, thereby ensuring everyone had a voice. In the second meeting it was made clear to participants that AV and JK would not speak during the interview component.”

Results: The results are exhaustive. Almost, I feel the results can be better represented using tables showing codes, themes, and sub-themes rather than a 12-page summary of the results. Quotations should not overwhelm your narrative. The bulk of your results section should be your interpretation and thematic synthesis, with quotations serving as evidence.

Whilst we appreciate this suggestion, given reviewer 1’s comments we have chosen to leave the quotations as currently presented.

Section-wise Comments

Abstract: The research questions can be removed from the abstract, and the authors can briefly mention the aims and objectives of the study. A lot of points in the abstract six themes labelled 1.,2., 3., … and then five principles…1,2,3…

The abstract has been revised and the research questions removed. Instead the aims of the study have been described:

“

---

## [Editor Report · Decision Letter 1]

29 Oct 2025

Dementia is our “biggest expanding caseload”: Core learning for student speech and language therapists

PONE-D-25-30844R1

Dear Dr. Volkmer,

We’re pleased to inform you that your manuscript has been judged scientifically suitable for publication and will be formally accepted for publication once it meets all outstanding technical requirements.

Kind regards,

Rohit Ravi, Ph.D.

Academic Editor

PLOS ONE

Additional Editor Comments (optional):

The revision is satisfactory!
---

## [Editor Report · Acceptance letter]

PONE-D-25-30844R1

PLOS ONE

Dear Dr. Volkmer,

I'm pleased to inform you that your manuscript has been deemed suitable for publication in PLOS ONE. Congratulations! Your manuscript is now being handed over to our production team.

Kind regards,

on behalf of

Dr. Rohit Ravi

Academic Editor

PLOS ONE